# Time-varying drainage basin development and erosion on volcanic edifices

Daniel O'Hara[1,2], Liran Goren[3], Roos M.J. van Wees[1], Benjamin Campforts[4], Pablo Grosse[5,6], Pierre Lahitte[7], Gabor Kereszturi[8], Matthieu Kervyn[1]

[1]Department of Geography, Vrije Universiteit Brussel, Pleinlaan 2, 1050 Elsene.
[2]Helmholz Center Potsdam, GFZ German Research Center for Geosciences, Potsdam, Germany.
[3] Ben Gurion University of the Negev, Department of Earth and Environmental Sciences, Beer-Sheva, Israel
[4] Institute of Arctic and Alpine Research, University of Colorado Boulder, Boulder, CO, USA
[5] Consejo Nacional de Investigaciones Científicas y Técnicas (CONICET), Argentina
[6] Fundación Miguel Lillo, Miguel Lillo 251, (4000) Tucumán, Argentina
[7] Université Paris-Saclay, CNRS, Laboratoire GEOPS, Rue du Belvédère, 91405 Orsay, France
[8] Volcanic Risk Solutions, School of Agriculture and Environment, Massey University, 4474, New Zealand

*Correspondence to*: Daniel O'Hara (Daniel.OHara@vub.be)

**Abstract.** The erosional state of a landscape is often assessed through a series of metrics that quantify the morphology of drainage basins and divides. Such metrics have been well-explored in tectonically-active environments to evaluate the role of different processes in sculpting topography, yet relatively few works have applied these analyses to radial landforms such as volcanoes. We quantify drainage basin geometries on volcanic edifices of varying ages using common metrics (e.g., Hack's Law, drainage density, number of basins that reach the edifice summit, as well as basin hypsometry integral, length, width, relief, and average topographic slope). Relating these measurements to the log-mean age of activity for each edifice, we find that drainage density, basin hypsometry, basin length, and basin width quantify the degree of erosional maturity for these landforms. We also explore edifice drainage basin growth and competition by conducting a divide mobility analysis on the volcanoes, finding that young volcanoes are characterized by nearly-uniform fluvial basins within unstable configurations that are more prone to divide migration. As basins on young volcanoes erode, they become less uniform but adapt to a more stable configuration with less divide migration. Finally, we analyze basin spatial geometries and outlet spacing on edifices, discovering an evolution in radial basin configurations that differ from typical linear mountain ranges. From these, we present a novel conceptual model for edifice degradation that allows new interpretations of composite volcano histories and provides predictive quantities for edifice morphologic evolution.

## 1.0 Introduction

Understanding how drainage basins on eroding landforms develop and evolve is a fundamental principle of Geomorphology. Over regional scales, basin geometry, structure, and spacing evolve in response to both external (e.g., climate, tectonics; Castelltort et al., 2012; Duvall and Tucker, 2015; Han et al., 2015; Yang et al., 2015) and internal (e.g., channel piracy; Bishop, 1995; Whipple et al., 2016) forcing as topographic slopes adjust to develop and maintain an equilibrium between erosion and uplift (e.g., Willett et al., 2001; Castelltort et al., 2009). As these landscapes adjust, transient signals within basins propagate upstream to surrounding channel heads, where opposing signals between adjacent basins drive divide migration that modify available area for overland flow (e.g., Willett et al., 2014; O'Hara et al., 2019).

Work in the 20th century established foundational relationships between basin drainage areas, lengths, and slopes (e.g., Horton, 1945; Strahler, 1952; Hack, 1957; Flint, 1974), providing the basis for analyzing landscape disequilibrium and evolution in both tectonically-active (e.g., Kirby and Whipple, 2012; Fox et al., 2014) and passive (Prince and Spotila, 2013; Willett et al., 2014; Braun, 2018) regions. These relationships are built on the

assumption of a dominantly-dendritic fluvial network existing on a near-linear primary landform (e.g., a mountain
range; Castelltort and Simpson, 2006). Furthermore, basin competition is often considered in the simplified
configuration of a binary drainage system, where a divide supports only two opposing basins that compete across it
(e.g., Gilbert, 1909; Mudd and Furbish, 2007).
Although dendritic channel networks are most prevalent on Earth, they are not the only type of configuration.
Trellis, rectangular, parallel, and radial drainages also occur (Howard, 1967). The formation of these other drainages
often relate to the region's tectonic, volcanic, or glacial history, subsurface structure, or geometry of the primary
landform that they erode (Zernitz, 1932). However, compared to dendritic basins, studies that explore the geometries
and evolution of other drainage settings are scarce (e.g., Mejía and Niemann, 2008; Becerril et al., 2021; Hamawi et
al., 2022).
Volcanic edifices are characterized by radial drainages. In these settings, quantifying drainage evolution can be
challenging as these landforms experience interspersed, short-term eruptive episodes superimposed onto the long-
term degradation record (e.g., Thouret et al., 2014). These stochastic volcanic events often produce spatially-varying
excess sediment supply in the form of pyroclasts with varying grain properties that significantly alter fluvial
transport on decadal scales (e.g., Major et al., 2018; Hayes et al., 2002). Additionally, drainage formation can lag
behind surfacing by volcanic deposits over 1 – 100 kyr timescales due to transmission losses associated with
permeable volcanic material (e.g., lava flows, pyroclasts; Lohse and Dietrich, 2005; Jefferson et al., 2010; Sweeney
and Roering, 2017). Finally, the more symmetric drainage divide configuration typical of linear mountain ranges
breaks down on volcanic edifices due to their radial nature, with multiple catchments constrained to the conical
structure of the volcano and converging towards one or a few main summits. Despite these challenges, volcanic
edifices represent ideal primary landforms to investigate drainage evolution due to their well-defined conical initial
conditions, datable surfaces, and scarce inheritance from regional tectonics. Furthermore, quantifying the
relationships between edifice construction and drainage basin morphology provides new insight for investigating
edifices remotely, and can thus expand our understanding of basin dynamics while also complementing field-based
surveys to resolve volcano edifice histories.
Here, we explore the development of drainage basins and topography on stratovolcanoes from Indonesia, Papua
New Guinea, New Zealand, and Guatemala (Fig. 1). Using common hydrographic metrics and broad volcanic
histories, we determine stages of maturation during basin evolution and derive a new generalized model for
stratovolcano degradation that builds off of previous studies (Ollier, 1988). We then quantify divide mobility on
radial structures within the context of our conceptual model and discuss the applicability of our analyses to
characterize an edifice's history.

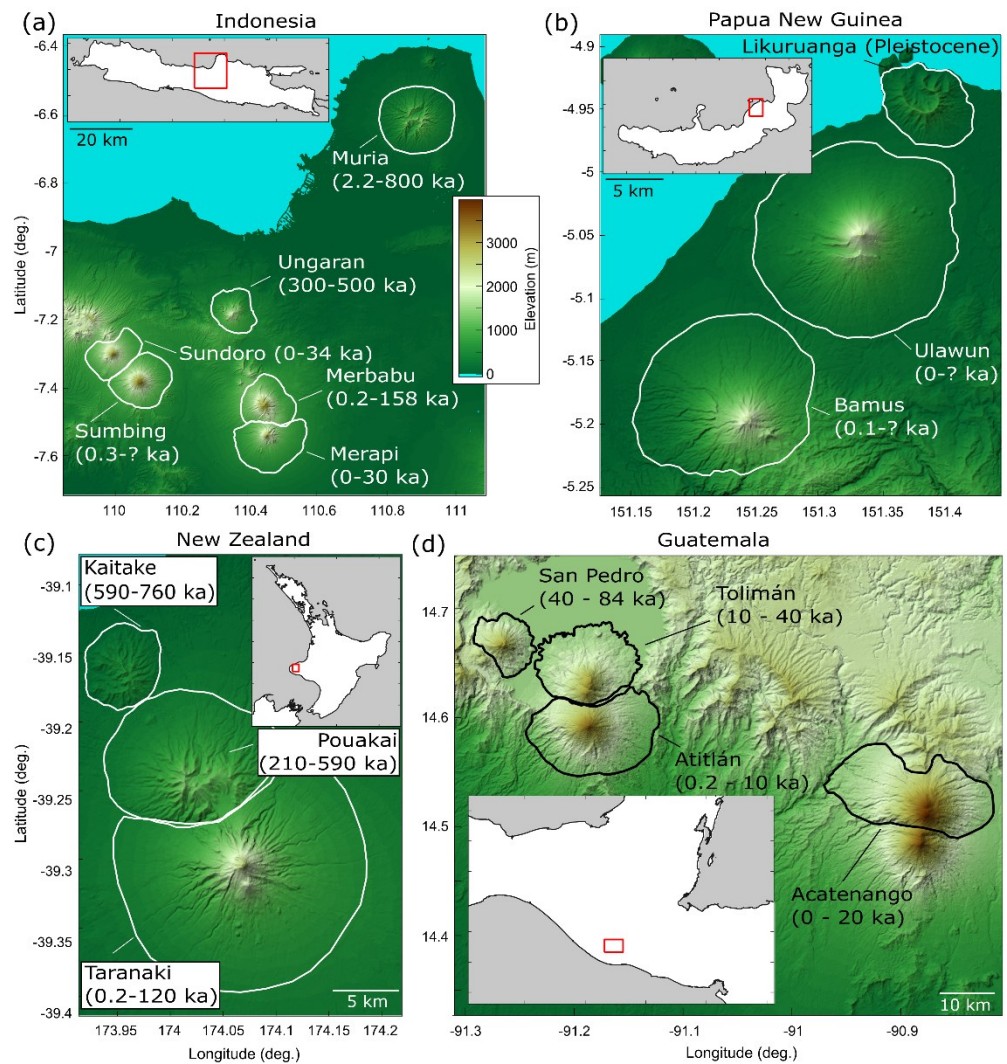


**Figure 1** – Regional hillshaded relief maps of 16 analyzed edifices from (a) Indonesia, (b) Papua New Guinea, (c) New Zealand, and (d) Guatemala. Maps are projected to 30 m UTM and use the same color scale. Solid white lines in a-c and solid black lines in d represent edifice boundaries (boundary definition described in Methods). Text describes volcano names and known ages of activity (Table T2). Insets are larger-scale regional maps for reference; gray areas represent ocean, white areas are land, and red squares are bounds of hillshaded maps.

## 2.0  Methods

To constrain the temporal evolution of stratovolcano morphologies, we focus on closely-spaced volcano sets (Fig. 1).  The advantages of this approach are that within each respective region, 1) volcanoes were likely fed by similar magma sources (e.g., Locke and Cassidy, 1997; Haapala et al., 2005; Mulyaningsih and Shaban, 2020), constructed by similar volcanic deposits, and thus had similar volcanic shapes, 2) edifices experienced similar climate conditions, 3) volcano sets have radiometric ages related to their initiation and most recent eruption that are comparable, providing constraints on their overall lifespan, and 4) volcanoes within the same set were active over different time intervals, thus showing contrasting time-dependent degrees of dismantling within a short (10's of km) distance. In order to consider drainage basin evolution through fluvial erosion from the perspective of radial landforms, we exclude volcano massifs from our analysis, as well as any volcano with recognizable collapse scars,

and only consider volcanoes that do not have an extensive glacial history. All analyzed volcanoes are classified as
stratovolcanoes by the Smithsonian Global Volcanism Program (Global Volcanism Program, 2013).

**2.1    Edifice Delineation**

Although automated algorithms exist to generate volcano edifice boundaries (e.g., Bohnenstiehl et al., 2012;
Euillades et al., 2013), these often create conservative limits around the edifice that ignore lower flanks and volcano-
sedimentary aprons (e.g., O'Hara et al., 2020). We thus follow the method suggested by van Wees et al. (2021) to
delineate edifice boundaries from surrounding topography. Using 30-m Shuttle Radar Topography Mission (SRTM)
Digital Elevation Models (DEMs) (Farr et al., 2007), projected in Universal Transverse Mercator (UTM) with
World Geodetic System (WGS 84), we first generate hillshade, aspect, and local slope rasters of the raw topography.
Lower edifice flanks are generally characterized by slope angles greater than some threshold value (Karátson et al.,
2012); we therefore remove short-wavelength variations of the slope raster by filtering it over a 300 m wavelength
(O'Hara et al., 2020) and contour regions that surpass a 3° slope threshold (van Wees et al., 2021). Using these maps
as visual aids, we then hand-draw boundaries that separate the edifice from surrounding terrain. Afterwards, the
DEMs are clipped using these boundaries to isolate the edifices for morphometric analysis. The planform areas of
edifice boundaries derived using this method range from 30.2 km$^2$ (Kaitake, New Zealand) to 432.7 km$^2$ (Muria,
Indonesia).

**2.2    Edifice Basin Morphology**

We analyze edifice basin morphologies with DrainageVolc, a series of scripts modified from TopoToolbox
(Schwanghart and Scherler, 2014), which is designed to investigate volcanic topography through a set of
topography-, drainage-, and channel-based analyses. The metrics considered here are commonly used within
tectonic settings but have not previously been applied to radial drainages. Figure 2 displays an example of our
methods using Ungaran volcano in Indonesia.
We first fill sinks in the DEM through TopoToolbox's preprocessing algorithm (Schwanghart and Scherler, 2014) to
ensure continuous flow to the edifice boundary and extract drainage basins from topography using steepest-descent
flow routing (Fig. 2a). We then perform a series of analyses related to basin geometry. The lengths ($L$) of all basins
draining to the edifice boundaries are calculated by determining mid-point paths between basin divides
perpendicular to the Euclidean distance between the highest and lowest reaches of the basin, irrespective of whether
there is an actual flow channel in this path (Fig. 2d). Assuming basins with total drainage areas ($A$) greater than
some threshold ($A_T$) support overland flow, we explore the correlation between the lengths and drainage areas of
these basins through a power-law regression to derive the Hack's Law relationship (Fig. 2b) for the edifice as
$$L = k_a A^H,$$  (1)
where $k_a$ and $H$ are Hack's coefficient and exponent, respectively (Hack, 1957). $H$ values are compared across
edifices as this exponent describes general basin geometry, with values of ~0.47 – 0.6 typically attributed to
dendritic systems (Hack, 1957; Mueller, 1972). Our Hack's Law derivation uses basin lengths as opposed to typical
flow path lengths to remove the effects of channel sinuosity and focus explicitly on basin geometry; however, within
the context of our edifice basins, this derivation does not significantly alter our results, and values are thus
comparable to those of previous studies (Fig. S1). We also analyze the density of the edifice's channel network by
extracting flow paths with drainage areas greater than $A_T$ from the landform, and calculate the edifice-scale drainage
density as
$$DD = \frac{\sum L_c}{A_E},$$    (2)
where $\sum L_c$ is the cumulative sum of all channel lengths and $A_E$ is the planform area of the edifice's boundary (Fig.
2a) (Horton, 1945). Using an automated slope-area analysis of basins to determine the drainage area threshold that
best corresponds with the power-law decrease in slope (Montgomery and Dietrich, 1994) for each edifice
(Supplemental text; Fig. S2), we find $A_T$ ranges between $0.32 - 1.62$ km$^2$, with a mean threshold of $0.85$ km$^2$ (Table
T1). For consistency across all edifices, we assume a constant drainage area threshold of $1.0$ km$^2$ to delineate
networks. Sensitivity analysis (Fig. S3) demonstrates that although the selection of $A_T$ does not significantly impact
the general behavior of drainage density results, Hack's Law exponent is more sensitive to this choice.

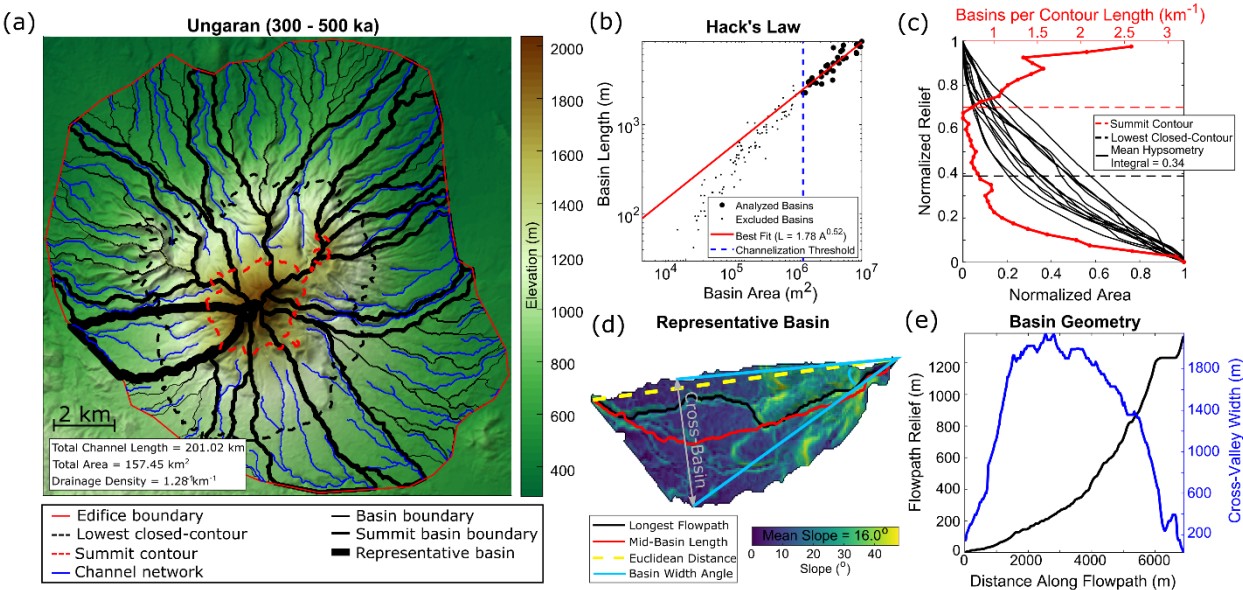


**Figure 2** – Analyzed basin metrics. **a:** Example from the map of Ungaran volcano (Indonesia), colored lines defined in the
legend. **b:** Hack's Law relationship between basin areas and lengths. Black circles are basins used in the power-law analysis,
black dots are excluded basins; blue-dashed line is the drainage area threshold ($A_T$; 1.0 km$^2$) for channelization. **c:** Scaled edifice
metrics. Red line shows normalized number of basins along elevation contours. Black lines are summit basin hypsometry curves.
**d:** Local slope and geometry values of representative basin (thick black line in 2a). Gray double-arrow represents cross-basin
direction (i.e., the extent of the basin) perpendicular to the Euclidean basin length. **e:** Cross-basin values along basin shown in 2d.
Black line is relief along the flowpath, blue line is cross-valley width.
Afterwards, we calculate mean values of basin geometries on each edifice. Rather than analyze the geometry of all
basins that exist on a volcano, we limit our analysis to larger basins that best characterize the edifice's drainage, and
thus its dismantling. These large characteristic basins may be determined using a variety of methods, such as
through an arbitrary number or percentage of basin sizes, using the basins that are within some radial distance of the
edifice's peak, or determining basins that extend to some portion of the edifice's height. Determining characteristic
basins by an arbitrary number or percentage of basin sizes may introduce bias as the population of basins drastically
varies between edifices (Fig. 8a), whereas determining characteristic basins by radial distance from the edifice's
peak introduces geometric constraints as edifice shapes often deviate from the textbook symmetric, single-peaked
edifice, instead developing large, irregular summit regions that are defined by high topography and multiple peaks
(e.g., Karátson et al., 1999; Grosse et al., 2012). As slope (and thus elevation) is an essential component of erosion
and basin development (Hack, 1957; Flint, 1974), we define characteristic basins as those that reach the edifice's
summit region. However, we note that defining characteristic basins based on radial distance can produce different
trends (Fig. S4) and may be more appropriate for some of our analyzed metrics (Section 5.3).
Generating a series of elevation contours along the edifice at intervals of 2.5% of the edifice's relief, we calculate
the number of basins that intersect each contour, normalized by the contour's length (Fig. 2c, red line). For all
edifices, we define the edifice's summit as the upper 30% of the edifice's relief, and thus consider the basins that
reach this summit region (referred here as *summit basins*) as those that best characterize the edifice's drainage
development. We then determine summit basin numbers, mean basin slopes (Fig. 2d), basin lengths ($L_B$; Fig. 2d, red
line), basin reliefs (Fig. 2e, black line), and maximum cross-basin widths ($W_B$; Fig. 2e, blue line). To compare
values across edifices of varying sizes, summit basin numbers are normalized by the length of the summit contour
(Fig. 2c) and basin reliefs are normalized by the relief of the entire edifice. We also utilize the radial nature of
edifices to generate normalized values of basin length ($L_B'$) and width ($W_B'$) as
$$L_B' = \frac{L_B}{L_E},$$ (3)
and
$$W_B' = 2\tan^{-1}\left(\frac{W_B/2}{L_{W_B}}\right),$$ (4)
respectively, where $L_E$ is the edifice's effective radius, defined as the radius of the circle with the same planform
area ($A_E$) as the edifice's boundary ($L_E = \sqrt{A_E / \pi}$), and $L_{W_B}$ is the distance from the highest point within a basin to
where the basin is widest. $W_B'$ thus converts basin widths into an angle relative to the summit (Fig. 2d, light blue
lines). Mean values of these quantities are then calculated for each edifice.
We also calculate mean summit basin hypsometry integrals for each edifice (Strahler, 1952; Fig. 2c, black lines).
Individual basin hypsometry curves ($H_C$) are derived by counting the number of basin pixels $N_{P_B}$ at or above
normalized elevation values ($\dot{Z}$, ranging from 0 to 1); afterwards, these values are normalized by the total number of
basin pixels ($N_{P_{Tot}}$) as
$$H_C(\dot{Z}_I) = \frac{N_{P_B}(\dot{Z} \geq \dot{Z}_I)}{N_{P_{Tot}}},$$ (5)
where $I$ is a counter over normalized elevation values from 0 to 1. Hypsometry integrals of each basin are calculated
as the positive integration over the curves from eq. (5). These are also averaged for each edifice.

## 2.3 Edifice Landform Morphology

As well as studying the temporal evolution of drainages on edifices, we also consider the broad geometry of the volcanoes. Grosse et al. (2009, 2012) developed the initial MorVolc algorithm in IDL, which quantifies edifice morphologies through a series of size, shape, slope, orientation, peak, and summit parameters. Using the same framework as DrainageVolc, we redeveloped the IDL code in Matlab, also utilizing the TopoToolbox DEM analysis package (Schwanghart and Scherler, 2014). Both DrainageVolc and the updated MorVolc scripts are available for use on GitHub (https://github.com/danjohara/Volc_Packages).

We analyze simple edifice geometry measurements with this updated version of MorVolc, including effective radius, height, height-radius ratio, and mean slope of the main flank (edifice region between the lowest closed-contour that encompasses the edifice and the summit contour, Fig. 2a). We also quantify the mean contour ellipticity and irregularity indices of the main flank from the previously-computed contours. The ellipticity index ($EI$) describes the elliptical nature of the edifice elevation contours, and is defined as

$$EI = \frac{\pi(L_M/2)^2}{A_C},$$ (6)

where $L_M$ is the length of the major axis of a best-fitting ellipse through the contour and $A_C$ is the area enclosed by the contour (Grosse et al., 2012). The irregularity index ($II$) describes divergence of the contour from a smooth ellipse as

$$II = di_{contour}\left(di_{ellipse} - 1\right),$$ (7)

where $di$ is the dissection index, defined as

$$di = \frac{P_C}{2A_C}\sqrt{A_C/\pi},$$ (8)

with $P_C$ and $A_C$ being the perimeter and area of the contour, respectively (Grosse et al., 2012). Finally, we also incorporate new measurements within MorVolc, including the slope variance of the entire edifice (standard deviation of all slope values divided by the mean slope, similar to roughness), as well as a minimum eroded volume estimate. Eroded volume is estimated from a convex-hull reconstruction of the edifice, using the methodology described in O'Hara and Karlstrom (2023), in which the footprints of individual elevation contours along the edifice are altered to remove concave regions (assuming they represent incised topography), thus creating convex polygons. Polygons are then interpolated in three dimensions to create a simplified, reconstructed edifice. Afterwards, the current topography is subtracted from the reconstructed edifice and positive values (i.e., areas having been eroded) are integrated to estimate the volume of eroded material. Finally, eroded volume is normalized as a percent relative to the total reconstructed volume.

Edifice landform and basin metrics that are based on average values (main flank mean slope, mean contour irregularity index, mean contour ellipticity index, as well as mean summit basin hypsometry, length, width, relief, and slope) have standard deviations of the sampled population that are presented as vertical bars in Figs. 3 – 4. Other

metrics (edifice height, radius, height-radius ratio, slope variance, normalized eroded volume, Hack's Law exponent,
drainage density, and normalized number of summit basins) are singular values for each edifice and thus do not have
associated standard deviations. Potential deviations of these values relate to the edifice's boundary, summit
designation, DEM source, or imposed drainage area threshold (Grosse et al., 2012; O'Hara et al., 2020; van Wees et
al., in review; Supplemental text).

**2.4     Edifice Ages**

To explore morphological evolution through time, we correlate edifice landform and drainage basin metrics to
volcano ages of activity. We thus compile known eruption records of each volcano, with ages ranging from present
to early Pleistocene (Table T2). Volcanoes often have complex surface evolutions, with lifespans of activity that
range 100-1000 kyrs and characterized by episodes of stochastic growth interspersed with periods of erosion during
quiescence (e.g., Karátson et al., 1999; Lahitte et al., 2012). Furthermore, episodes of activity are often constrained
to localized regions of the edifice and thus do not fully resurface the entire landform (e.g., Civico et al., 2022).
Similarly, erosion across the edifice is typically non-uniform as local conditions are dependent on the age and type
of activity, as well as microclimates (e.g., Ferrier et al., 2013; Pierson and Major, 2014; Thouret et al., 2014; Ricci et
al., 2015).
Despite the spatial and temporal heterogeneities of activity and erosion, we argue that a generalized morphologic
age of an edifice may be derived that quantifies the erosional state of the landform and relates to the edifice's
lithologic age. To account for the time differences between short-term events and the cumulative long-term history
on morphology, we define an edifice's age as a single value using the log-mean between the most recent eruption
and oldest date of activity. This definition thus accounts for the span of temporal magnitudes; however, we note that
using linear-mean ages produce similar results (Fig. S5) and recognize that other definitions of an edifice's
morphologic age are plausible (e.g., the time since the last eruption; Fig. S6). Afterwards, we analyze the temporal
evolution of edifice morphologies by fitting logarithmic relationships between edifice age and morphometric
parameters. Some volcanoes (Sumbing, Bamus, and Ulawun) have poorly-documented histories (only the most
recent eruption has been dated) and are therefore excluded from the regression. Conversely, Likuruanga is known to
have erupted only during the Pleistocene and is incorporated in the analysis.

**3.0  Results**

We find trends between stratovolcano age and our morphometry metrics through time (Figs. 3-4; Supplemental
Table T3). Considering all metrics, we find that edifice height, mean ellipticity index, normalized eroded volume,
Hack's Law exponent, drainage density, mean summit basin hypsometry integral, normalized basin length, and
normalized basin width have $R^2$ values ranging $0.39 - 0.77$ and correlation p-values $\leq 0.05$. This list expands to
include effective edifice radius and mean irregularity index by removing a notable outlier (Muria, Indonesia; Fig. 4b,
4e), suggesting all of these metrics provide quantitative measures to characterize the overall maturity of the edifice.
Other metrics have weaker correlation values $(0 - 0.25)$ and are statistically insignificant (p-values $> 0.1$), and thus
may be more sensitive to the initial edifice geometry or other processes that alter edifice morphology, or that age is
not a significant factor for these metrics. Muria (the noted outlier for effective edifice radius and irregularity index),
has an extensive volcanic history (from ~ 800 ka to 2 ka; McBirney et al., 2003; Global Volcanism Program, 2013)
and a morphology characterized by two broad fluvial networks on opposite flanks that are deeply incised into the
landform and may be associated with breached craters or flank collapses (Fig. 1a), suggesting this edifice may not fit
into the simple, radial volcano expectation of our dataset. We also note that due to the geometries that Acatenango
and Atitlán share with their sister volcanoes (Fuego and Tolimán, respectively; Fig. 1d), and our imposed definition
of an edifice's main flank (region between the lowest closed-contour and upper 30% of the edifice's height),
irregularity and ellipticity values could not be derived for these volcanoes.
Of the statistically-significant metrics related to edifice drainage morphology, mean summit basin hypsometry
integral and normalized width increase through time, whereas Hack's Law exponent, drainage density, and mean
summit basin normalized length decrease (Fig. 3). Similarly, considering statistically-significant metrics related to
the edifice as a primary landform, mean irregularity index, mean ellipticity index, and convex-hull based eroded
volumes increase with age, while edifice height and effective radius decrease with age (Fig. 4).

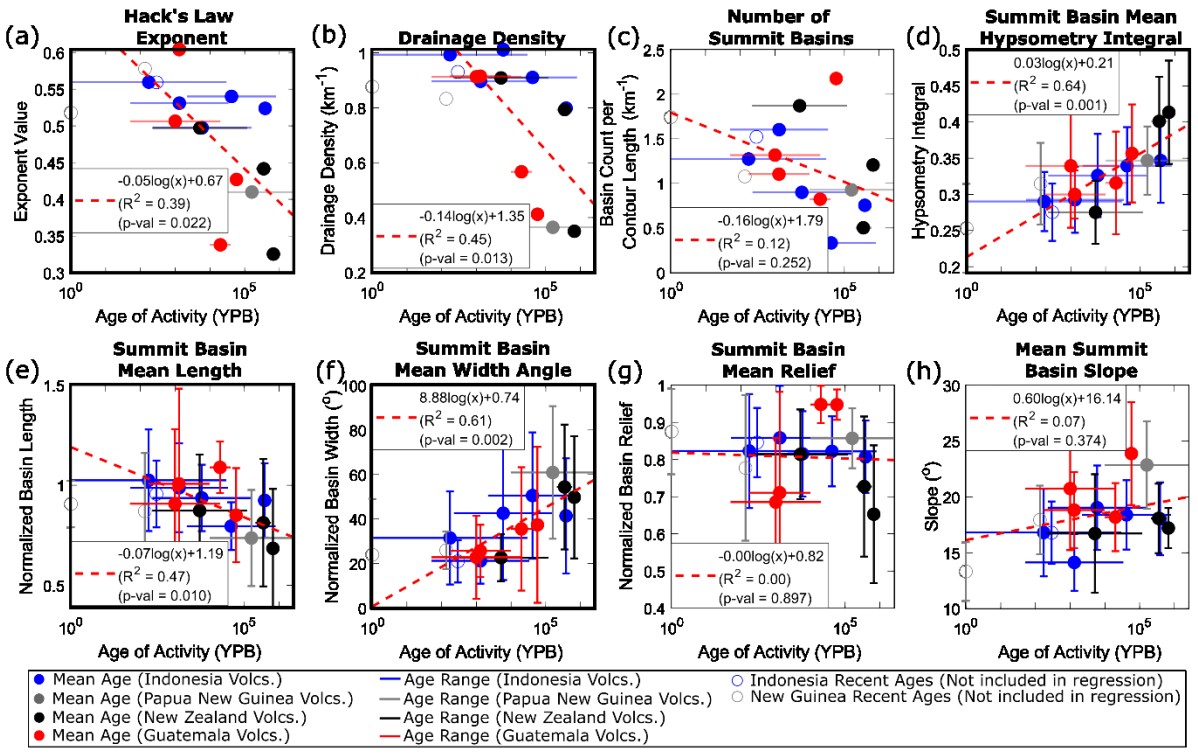


**Figure 3** – Temporal relationships of drainage basin morphology metrics. Colors correspond to volcanic region. Horizontal lines
are edifice age ranges of activity, with filled circles representing log-mean age. Vertical lines represent one standard deviations of
values (where appropriate). Red-dashed lines and equations characterize logarithmic regressions; open circles are excluded from
the regression due to age constraints. Thick black border highlights relationships with $R^2 > 0.35$.

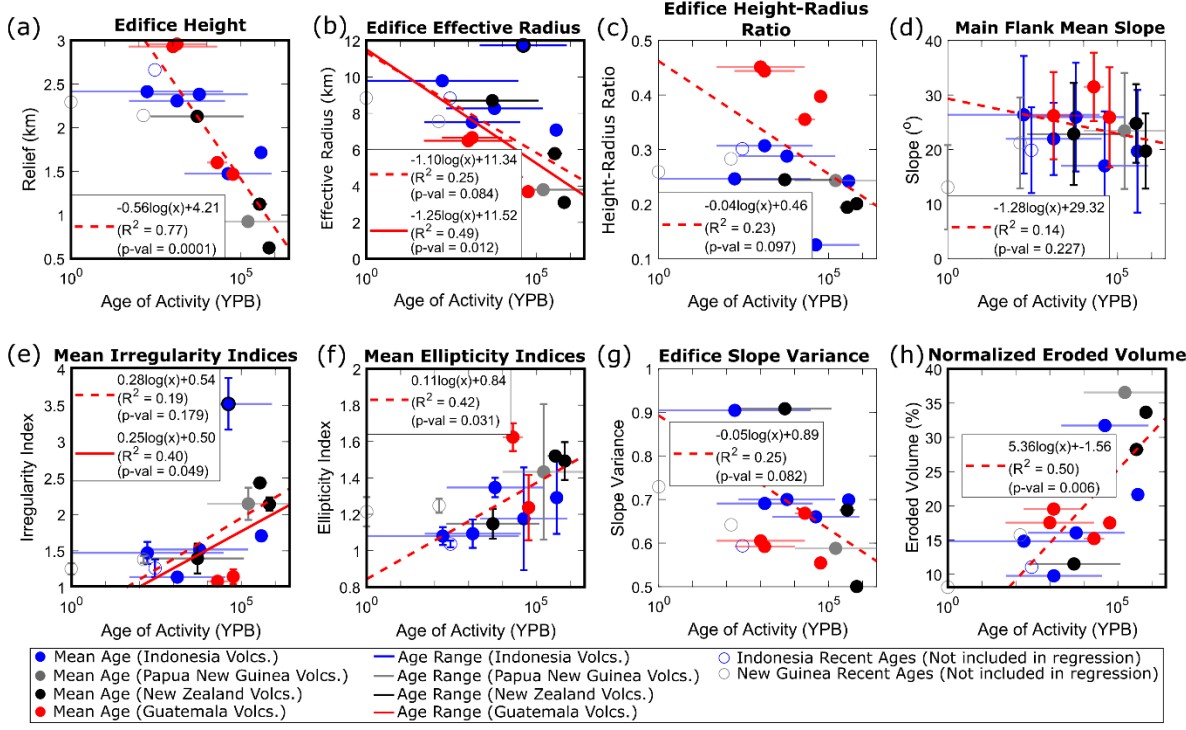


**Figure 4 –** Temporal relationships of landform morphology metrics. Colors and symbols are same as those described in Fig. 3.
Solid red lines in (b) and (e) are secondary regressions with outlier (Muria) excluded. Thick black border highlights relationships
with $R^2 > 0.35$.

## 4.0 Discussion

### 4.1 Generalized model for edifice degradation

The evolution of stratovolcanoes as primary landforms and the drainage basins that erode them are inextricably

linked. Our results thus establish a new framework for evaluating volcanic edifices by considering both the landform

and its drainage systems. This evolutionary model expands on stages previously defined qualitatively (Ollier, 1988)

and follows similar drainage evolution to that observed in badlands (Schumm, 1956).

Erosion of a stratovolcano can be described within the context of our metrics by considering a simplified, conical

edifice (Fig. 5). In the initial stages of erosion (Fig. 5a, equivalent to ~10% normalized eroded volume in Fig. 4h),

narrow (~ 20° normalized width angle) and uniform (normalized mean length near 1) drainages form that extend

from the summit region to the lower flanks (i.e., 'parasol ribbing'; Ollier, 1988), giving a high drainage density (~1

km$^{-1}$) and Hack's Law exponent (~0.6).

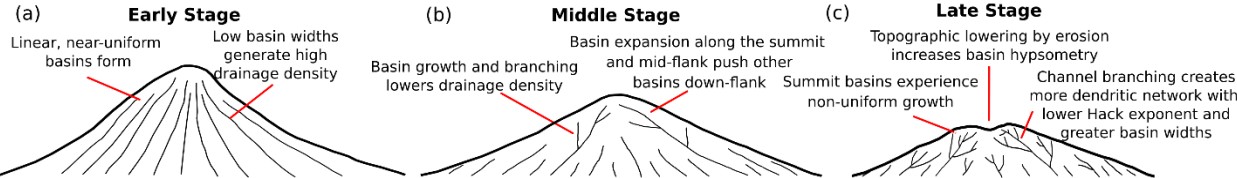


**Figure 5 –** Conceptual model of edifice dissection based on interpretation of temporal morphologic trends shown in Figs. 3 – 4.
Thin black lines represent drainage systems.


As the edifice degrades to 30-40% normalized eroded volume (Fig. 4h) on 10-100 kyr timescales (Fig. 5b-c), both
its height and area decrease; however, height decreases faster, leading to a decrease in height-radius ratios. The
erosion of the edifice is accompanied by drainage basin growth, with summit basins expanding azimuthally along
the edifice to normalized basin widths of 40-60°, pushing the headwaters of other basins down the edifice flanks.
Furthermore, as summit basins expand, they incise into the edifice flanks and develop a more dendritic structure
associated with lower drainage density (~0.5 km$^{-1}$) and Hack's Law exponent (~0.4). This is accompanied by non-
uniform summit basin growth that causes normalized basin lengths to decrease below 1.
As the edifice erodes, processes occur over varying scales to alter general edifice morphology: 1) over the entire
edifice, erosion-driven topographic lowering occurs faster than horizontal areal loss of the edifice, creating a flatter
landform; and 2) at the scale of a basin, incision carves into the initially-planar flanks of the edifice, steepening
surrounding valley walls and increasing contour irregularity. The relationship between basin-scale incision and
edifice-scale flattening is recorded through summit basin hypsometry integrals, with increasing values suggesting
that edifice-scale flattening is the dominant process. This leads to a scale-dependent behavior in edifice morphology
– although the edifice as a landform is becoming flatter, incision causes topography to steepen locally. Previous
studies (e.g., Karátson et al., 2012; Dibacto et al., 2020; Ollier, 1988) suggest this simultaneous behavior causes the
edifice to lose its conical, single-peaked nature over longer (> 1 Myr) timescales, developing high-relief drainage
divides over an extended summit region that support binary basin competition as the edifice erodes to the same relief
as the surrounding terrain. Furthermore, we note that the decrease in edifice area through time differs from the
expectation of a sedimentary apron around the edifice that increases in area as the edifice erodes. Since edifice
boundaries are consistently defined in-part by a 3° topographic slope threshold, this suggests that on the 100 kyr
scale, sediment is not depositing at the edifice's base, but is being evacuated from the vicinity of the edifice, likely
through fluvial transport. The loss of sedimentary apron and overall decrease in edifice planform area was also
suggested by Ollier (1988) as an edifice transitions from its 'intact' stage to 'planèzes' stage.
This conceptual model represents a generalized view of edifice degradation, as a variety of processes (both volcanic
and erosional) can impact an edifice's morphology throughout its lifespan. Furthermore, other climate conditions not
considered here (e.g., glaciers, arid environments) are expected to alter the patterns and rates of basin evolution.
Nonetheless, we propose that, barring major events that significantly alter topography, stratovolcano degradation by
fluvial processes generally follows the model presented here.
**4.2    How do basins compete on radial structures?**
Our results suggest that drainages on radial structures are highly dynamic. From initially-uniform basin geometries,
preferential erosion causes basins near the summit to become more dominant and expand, forcing other basins
down-flank and generating a 'topographic hierarchy', with higher-order basins spanning the entire flank of the
edifice and lower-order basins occurring on lower sections, analogous to inferred basin evolution on linear fault
blocks (Talling et al., 1997). This hierarchy of basin ordering is a direct product of non-uniform basin development
over the edifice that contributes to the preservation of less-eroded portions of the lower flanks (i.e., planèzes; Ollier,

319  1988).

Non-uniform basin development and transience is a natural component of landscape evolution (e.g., Hasbargen and
Paola, 2000); however, various factors (both volcanic and non-volcanic) can influence erosional patterns and
accentuate basin growth across volcanic edifices. These may include 1) local slope changes associated with
magmatic intrusions (e.g., Wicks et al., 2002; Biggs et al., 2010; Castro et al., 2016) or mass-wasting (e.g., Ui and
Glicken, 1986; Shea and van Wyk de Vries, 2008); 2) variable volcanic eruption activity that increase sediment
loads (Hayes et al., 2002; Pierson and Major, 2014), alter infiltration and rock erodibility (e.g., Wells et al., 1985;
Sklar and Dietrich, 2001; Jefferson et al., 2010), or remove bedrock through scouring by pyroclasts (Gase et al.,
2017) or melting by lava flows (i.e., thermal erosion; Kerr, 2001) during deposition; 3) non-uniform changes in
overland flow and stream power associated with breached craters (e.g,. Karátson et al., 1999) or edifice-scale
precipitation gradients (e.g., Ferrier et al., 2013); and 4) downstream alterations to drainage channels that migrate
upstream as a propagating incision wave (i.e., knickpoints; Kirby et al., 2003; Cook et al., 2013; Perron and Royden,
2013). The long-term compilation of such processes helps drive non-uniform erosion across the edifice, which in
turn encourages divide migrations and changes in basin size and geometry. More specifically, basins that exhibit
higher erosion rates would tend to expand at the expense of their neighboring basins and potentially become the
dominant basins, while lower erosion rates will cause other basins to shrink and their boundaries to migrate further
down the edifice's flank.
The morphology of drainage divides is sensitive to differences in erosion between neighboring basins and can thus
be used to characterize basin competition. We quantify basin geometry unsteadiness through an exploration of
divide stability using the *divide asymmetry index* (*DAI*; Forte and Whipple, 2018; Scherler and Schwanghart, 2020),
calculated as the positive difference in hillslope relief (vertical distance between the ridge and nearest channel)
across a divide and normalized by the sum of hillslope reliefs, ranging between 0 (symmetric) and 1 (asymmetric).
We limit our analysis to only consider divides that correspond to fluvial basins (i.e., have drainage areas > 1.0 km$^2$
(Scherler and Schwanghart, 2020).
Divide mobility is expressed using probability density functions (PDFs) of *DAI* for all volcanoes (Fig. 6a). A clear
temporal trend emerges – older volcanoes have larger distributions clustered around lower (< 0.4) *DAI* that rapidly
decrease with increasing *DAI*; while younger volcanoes show monotonically-decreasing distributions, with fewer
normalized populations of low-*DAI* and greater normalized populations of high-*DAI* values compared to older
volcanoes. Integrating these PDFs into single values (referred to here as *Γ*; Fig. 6b) shows a moderate correlation
with age (R$^2$ = 0.38) with the removal of Likuruanga (Papau New Guinea) as an outlier, which may be associated
with a breached crater (Fig. 1b).
Combined with basin morphology trends (Fig. 3), this suggests younger volcanoes have basins with more uniform
planform geometries and less-stable basin configurations. As the edifice erodes, basin planform geometries become
less uniform, but develop more stable configurations as evidenced by the greater symmetry of hillslope relief across
divides. The relationship between basin non-uniformity and stability can be observed spatially by comparing *DAI*
values between Merapi (youngest) and Kaitake (oldest) volcanoes (Fig. 6c-d). Highest *DAI* values on both
volcanoes generally occur at the mid- and lower-flanks of the volcano, suggesting basin expansion occurs mainly
azimuthally along edifice flanks, rather than across the edifice summit. This spatial analysis highlights the process
that generates topographic hierarchy – by expanding azimuthally, basin growth drives less-dominant basins down-
flank through a zippering process, creating drainages with tapered geometries along the lower flanks.

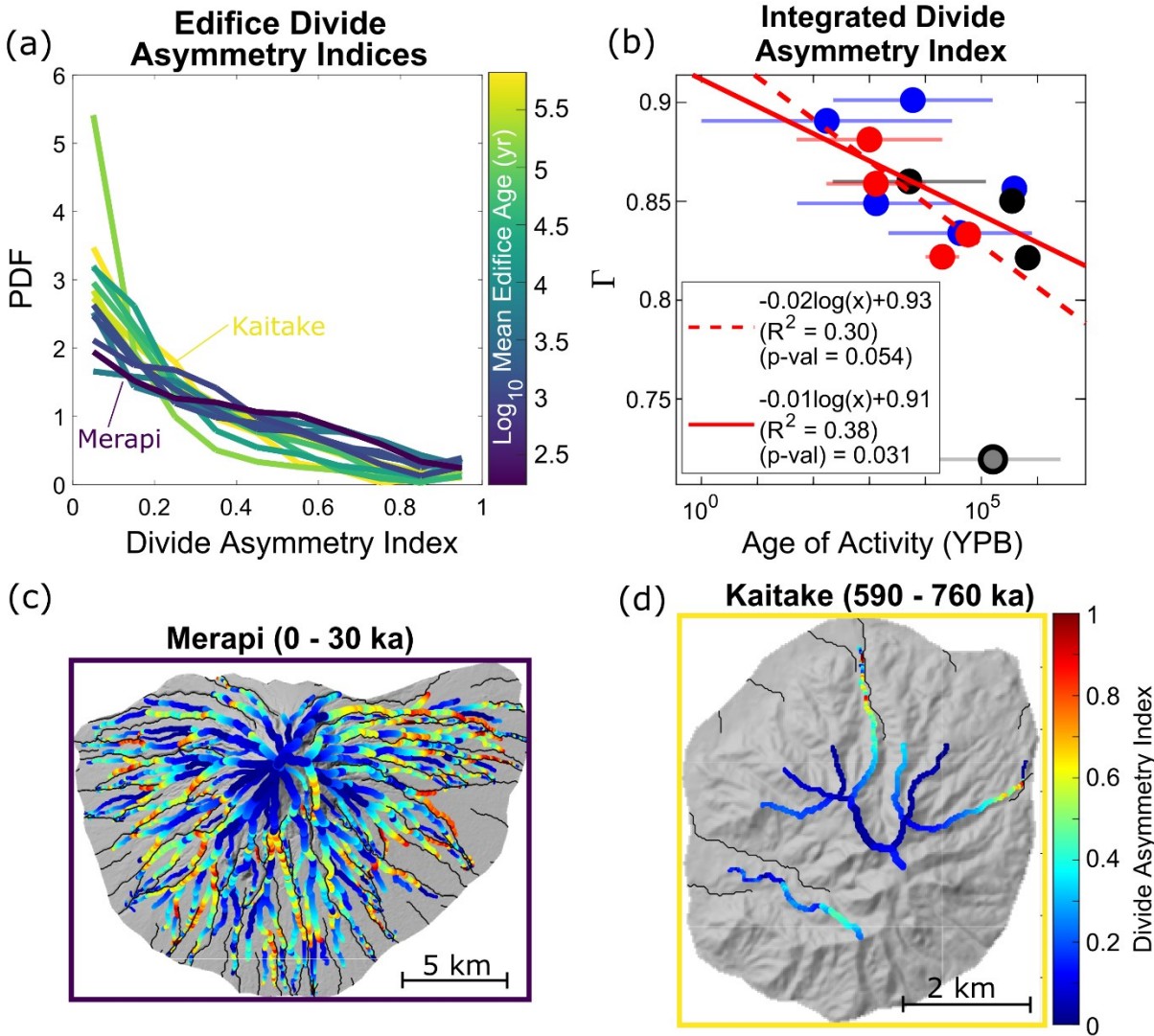


**Figure 6 – a:** Probability density functions (PDFs) of volcano divide asymmetry indices (*DAI*); colors correspond to log-mean
edifice ages. **b:** Integral of PDFs (*Γ*) compared to edifice age. Colors and symbols are the same as Fig. 4. **c-d:** *DAI* values for (c)
Merapi (Fig. 1a) and (d) Kaitake (Fig. 1c) at the divides. Background images are hillshade of topography, black lines are edifice
channel network. Borders are colored with respect to Fig. 6a color scale.
**4.3    Edifice basin widths and spacing**
Our results show that edifices experience the same morphologic trends when considering the number of basins along
edifice relief (Fig. 7a): lower flanks are characterized by normalized basin numbers between 2–5 km$^{-1}$, main flanks
are characterized by relatively consistent normalized basin numbers < 2 km$^{-1}$, while the normalized basin numbers

increase near the summit (upper 30% of the edifice). This trend appears to occur largely independent of age, even within the upper flank (as demonstrated by a low $R^2$ value of 0.12 at the summit contour, Fig. 3c), suggesting that this morphologic trend is a direct consequence of the conical nature of volcanoes. Furthermore, non-normalized summit basin numbers also demonstrate a weak temporal trend, both at the upper 30% height designation (Fig. 7b) as well as other percentages (Fig. S7). This suggests that basins that initially form on the summit region may retain their topographic position as the edifice erodes. However, Fig. 3f demonstrates that these basins still widen through time, to a width angle of ~60°, though further analysis on older volcanoes is needed to explore whether this persists on the Myr-timescale.

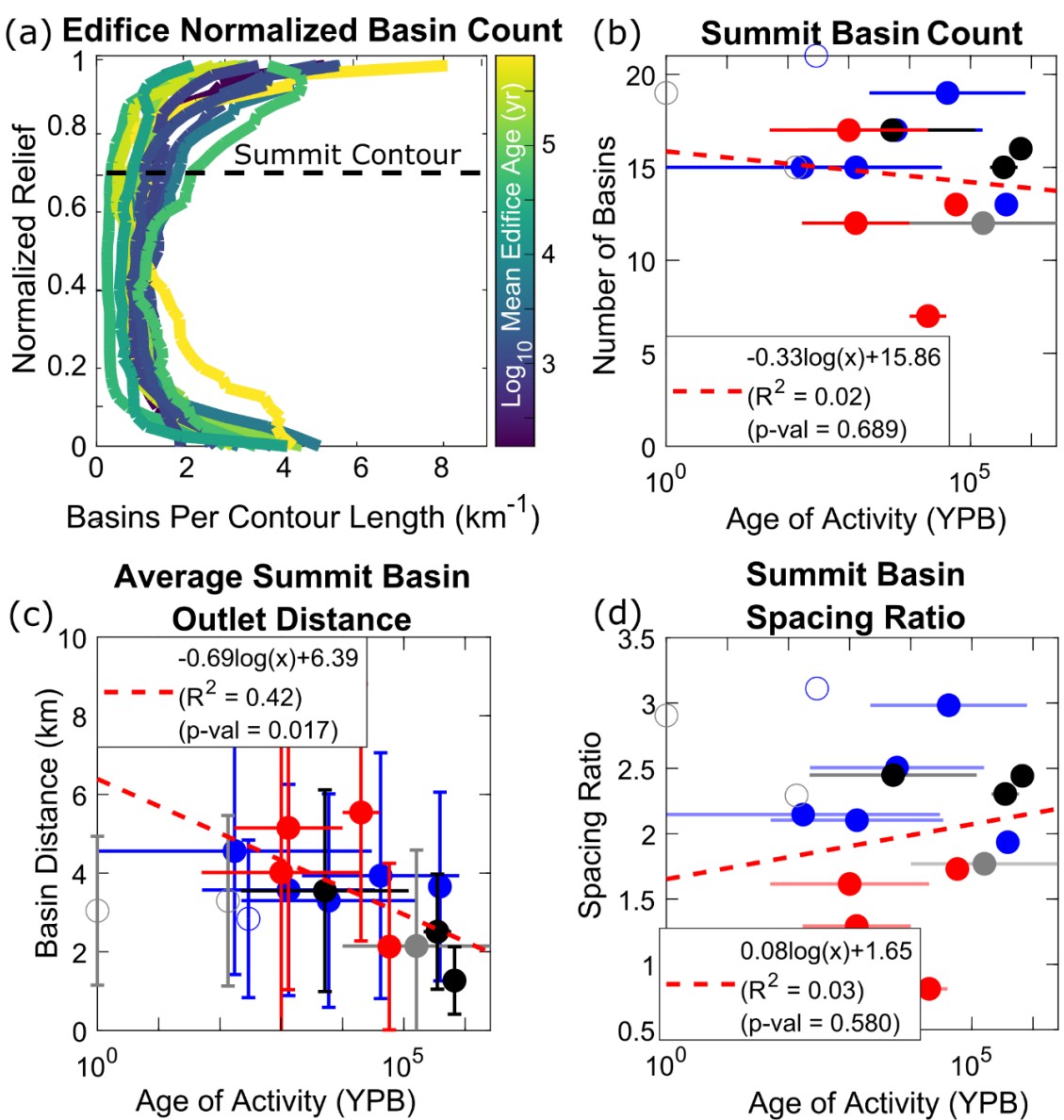

**Figure 7** – **a:** Normalized number of basins along normalized relief for each volcano; colors are log-mean edifice age. **b:** Non-normalized number of summit basins (defined by the upper 30% of the edifice's height; black-dashed line of a) compared to log-mean edifice age. **c:** Average along-perimeter summit basin distance compared to edifice age. **d:** Summit basin spacing ratio (data from Fig. 4b divided by data from c) compared to edifice age. Colors and symbols in b-d are the same as Fig. 3.

An apparent contradiction occurs when comparing mean summit basin width angles to the number of summit basins.
If all summit basins reached a width angle of ~60°, it would be expected that only ~6 basins would exist at the
summit; however, Fig. 7b shows that the number of basins that reach the summit on most edifices is greater than 10.
This difference is a consequence of radial drainage basins achieving their maximum widths at different heights
relative to the height of the edifice, such that basin widths are normalized by different distances from the summit.
Indeed, as discussed in Section 4.2, divide asymmetry is most frequent in the mid- and lower-flanks of the edifice
(Fig. 6), thus accommodating largest basin widths at different sections of the flank.
If the number of basins that reach the summit is time invariant, how does this translate to the circumferential spacing
of their outlets at the base of the edifice? Hovius (1996) compiled the ratio between mountain belt half-widths
(distance between the major divide and mountain front, $W_M$) and distances between major drainage basin outlets
(those that reach the major divide; $s$) in 11 mountain ranges globally, and determined a globally-averaged spacing
ratio ($W_M$ / $s$) of ~ 2-3. We perform a similar analysis by dividing edifice effective radii by the average along-
perimeter spacing between summit basin outlets. Figs 4b and 7c show that while edifice effective radii decrease
through time, so does the average perimeter distance between summit basin outlets. These behaviors thus combine
to produce summit basin spacing ratios of ~1 – 3 (Fig. 7d), consistent with Hovius (1996) as well as modeling
studies of drainage patterns (Habousha et al., 2023). This suggests that while summit basins azimuthally expand
their widths, the edifice is also decreasing in area as the landform erodes, thus decreasing the distances between
summit basin outlets.
However, a different behavior emerges when considering basins by their radial distance relative to the edifice's peak
(Fig. 8), which is more sensitive to the areal expansion of basins along the edifice's flank. Plotting the non-
normalized number of basins as a function of radial distance (normalized by maximum radius for each edifice) and
time shows a clear temporal trend (Fig. 8a), with younger edifices having more basins along all sections of the
volcano (as schematized in Fig. 5). This trend becomes more apparent through the logarithmic regression between
edifice age and the number of basins that exist at 30% radial distance from the peak (Fig. 8b), with other normalized
distances showing the same behavior (Fig. S8). Conducting a similar outlet perimeter-distance analysis on these
basins shows that the average distance between basin outlets is relatively constant at ~2 km (Fig. 8c), giving a
temporal decrease in basin spacing ratios ($R^2$ = 0.35, Fig. 8d). This relationship suggests a dynamic in radial
drainage evolution related to landform geometry. Combined with other metrics, our results suggest that as the
edifice erodes and loses planform area through time, very small basins on the edifice's lower flanks likely become
erased while more dominant basins widen on the mid flank, thus causing basins that exist within 30% radial distance
of the edifice's summit to retain an approximately constant outlet distance along the shrinking perimeter.

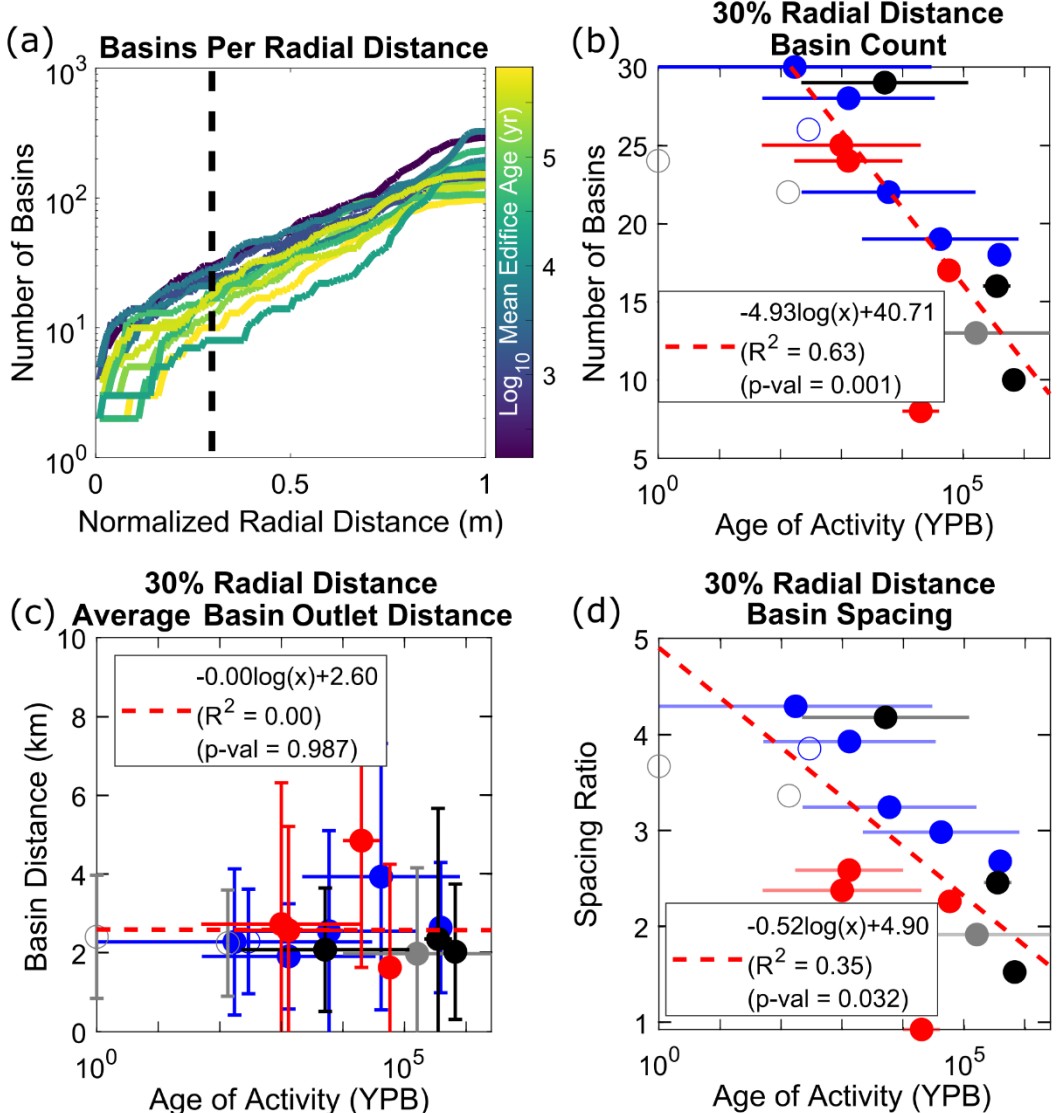

**Figure 8** – **a:** Non-normalized number of basins as a function of normalized distance from the edifice's peak; colors are log-mean edifice age, black-dashed line represents 30% normalized radial distance from the edifice's peak (basins used for plots in b-d). **b:** Non-normalized number of basins compared to log-mean edifice age. **c:** Average along-perimeter basin distance compared to edifice age. **d:** Basin spacing ratio (data from Fig. 4b divided by data from c) compared to edifice age. Colors and symbols in b-d are the same as Fig. 3.

## 4.4    Radial drainage basin area-length relationship

As a final observation for volcanic edifice drainage basins, we consider basin geometries in reference to Hack's power-law relationships between basin areas and lengths (Hack, 1957). Analyzing Hack's Law regressions for Merapi and Kaitake (Fig. 9), the relationships between spatial location and basin geometries become apparent. On Merapi, basins less than $10^5$ m$^2$ do not conform to the same power-law trend as those greater than $10^5$ m$^2$, whereas on Kaitake this break occurs at $10^6$ m$^2$. These smaller basins are constrained to the lowest regions of the edifices' flanks and likely correspond to non-channeled surfaces. Of those considered for the Hack's Law regression, the $\log_{10}$ basin length deviation ($D_L$) from the power-law is calculated as

$D_L = \log_{10}(L_H(A)) - \log_{10}(L),$             (9)
where $L_H$ is the basin length of the Hack's Law regression from a given basin's area ($A$), and $L$ is the basin's length.
As expected from the geometric relationship, basins that fall below the power-law regression ($D_L < 0$) are wider,
and those that are above the power-law regression ($D_L > 0$) are narrower.
Calculating $D_L$ for basins with areas greater than our imposed channelization threshold (1.0 km$^2$), one clear
observation is the presence of highly-elongated basins on Merapi that exist on the mid- to upper-flanks and have $D_L$
values > 0.15 (Fig. 9c). These basins appear wedged or pinched between larger basins and would be expected to not
have as much growth potential compared to their wider neighbors. Elongated basins also exist on Kaitake; however,
they do not have as high of a deviation (maximum $D_L \approx 0.1$; Fig. 9d). This may be a product of the lower number of
basins that exist on Kaitake, the overall lower amount of drainage area that Kaitake basins occupy, or an evolution
of basins towards more consistent patterns, thus decreasing the amount of variability from the power-law
relationship. On both Merapi and Kaitake, these elongated basins may further highlight the dynamics of basin
competition on radial structures – through drainage divide migration and areal loss (likely influenced by edifice-
scale sector collapses or regrowth events; Gertisser et al., 2023), less-erosive drainages become passive players to
more dominant basins and adopt non-standard geometries, becoming narrow, chute-like basins on the mid- and
upper-flanks.

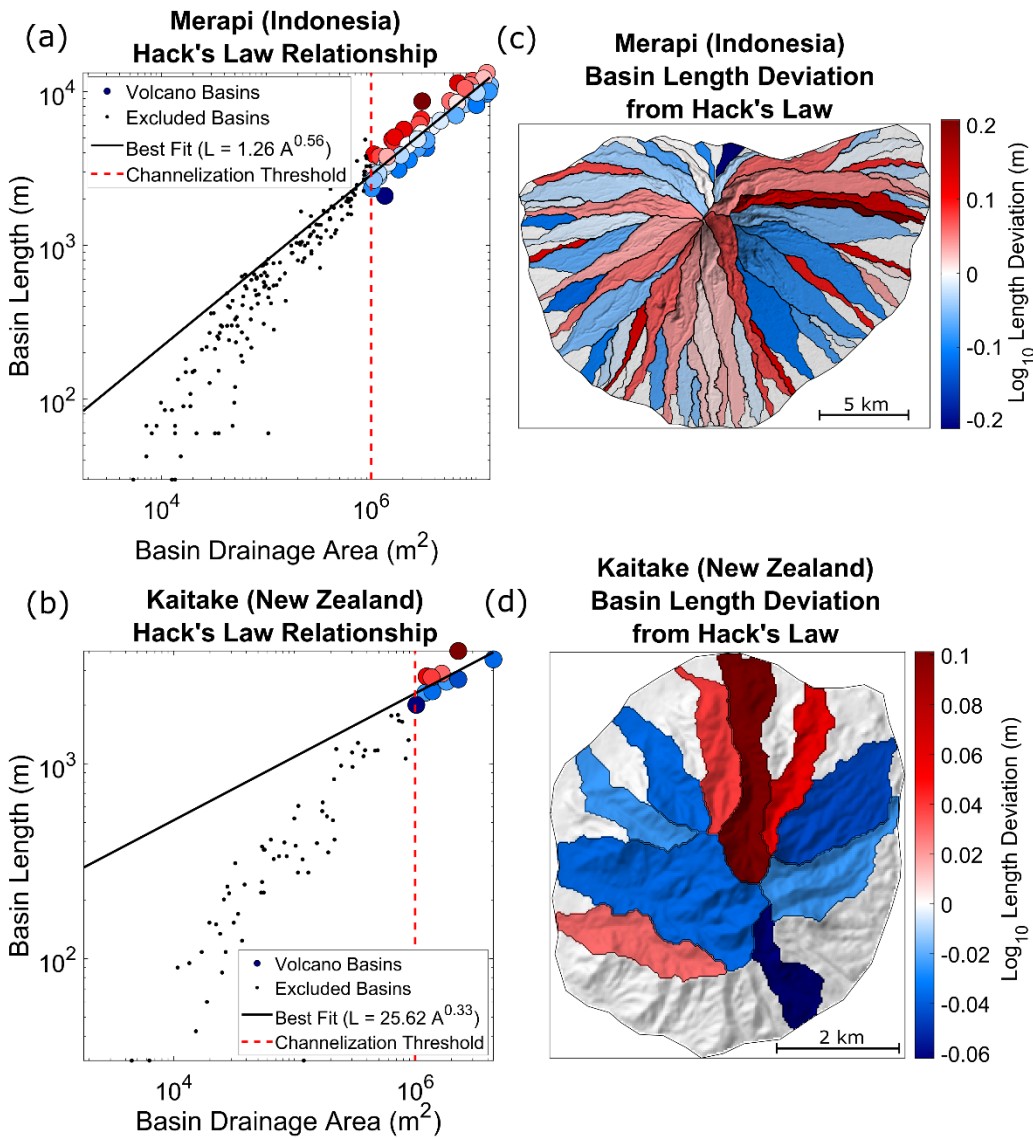

**Figure 9** – Hack's Law analysis of (a, c) Merapi and (b, d) Kaitake. **a-b:** Basin drainage area – length relationships. Black lines represent Hack's Law regressions. Colored circles correspond to the deviation from the regression trend (eq. 9), associated with the color bars in c and d. Red-dashed line is imposed 1.0 km² channelization threshold, black dots are basins less than the threshold and excluded from the regression. **c-d:** Semi-transparent hillshaded relief maps showing the deviation of each basin from the best-fit power-law regression.

## 4.5    How do radial drainages compare to other settings?

Thus far, our discussion has focused on deriving a foundational understanding of how radial drainages on volcanic

edifices evolve and compete. However, we note similarities between our interpretation and those from previous

studies in other drainage settings. This leads to a simple question – is there a significant difference between radial

and dendritic drainage development and evolution?

Our results show that basin formation on volcanic edifices follows the development of rills and gullies within

badlands (Schumm, 1956). As radial drainages evolve and certain basins expand to become dominant features on the

edifice, less-dominant basins become passive and are pushed down-flank, often adhering to non-standard geometries

as imposed by their more-dominant neighbors (Habousha et al., 2023; Beeson and McCoy, 2022). The dynamics of

this basin competition and formation of passive basins are demonstrated by edifice basin spacing ratios. Summit
basins on edifices have spacing ratios that appear time-independent and fit within the range of values observed in
linear mountain ranges globally (Hovius, 1996)  (Fig. 7), suggesting this ratio is set during the initial stages of basin
formation – an attribute of basin evolution that has been shown to occur on linear fault blocks (Talling et al., 1997;
Habousha et al., 2023). However, basins that are within a radial distance from the summit that is 30% of the
edifice's maximum radius do experience a temporally-decreasing spacing ratio and constant distance between
outlets (Fig. 8), capturing the development of a basin topographic hierarchy along the edifice – a behavior not
previously observed. Finally, our drainage divide analysis on volcanic edifices suggest that radial drainage basins
evolve towards a stable basin configuration as topography matures towards a dynamic equilibrium, similar to
regional landscape evolution globally (e.g., Perron and Royden, 2013; Willett et al., 2014).
This comparison suggests that drainage development and evolution on radial structures are largely similar to those
occurring within linear mountain settings. However, some differences still occur, particularly in relation to basin
geometries imposed by the larger-scale, radial primary landform. Dendritic drainages in linear mountain belts and
fault blocks are characterized by their leaf-like geometries (e.g., Zernitz, 1932; Strahler, 1952; Talling et al., 1997),
having a broad headwater region that decreases towards the outlet to a tapered point. Although radial drainages also
have tapered outlets and basin widths increase upstream, these widths are hindered by the conical geometry of the
primary landform and convergence of multiple basins towards the summit, leading to a tapered headwater as well as
a tapered outlet. This geometric constraint is well-demonstrated by the drainages on Merapi (Fig. 9c), where summit
basins are generally widest on the lower- or mid-flanks; however, this trend is not as obvious on Kaitake (Fig. 9d),
where erosion has dissected the landform and weakened the conical influence of the edifice on basin geometries.
Furthermore, as edifice drainages are limited to a conical landform, their evolution and configuration are constrained
by a cumulative areal limit. As opposed to linear mountain ranges (where a morphologic change in one basin
impacts its neighbors, which then impacts their neighbors as a cascading chain across the landscape), on volcanic
edifices, a morphologic change in one basin (particularly a dominant basin) may directly impact the erosional state
and morphology of most other basins on the landform due to the high number of basins that may share a divide with
this basin. This areal effect on radial basin evolution may be further augmented by the higher diversity of underlying
host rocks between edifice basins associated with magmatic and volcanic products (e.g., tephra deposits, lava flows,
and intrusions) that is not as prevalent within linear mountain ranges.
Despite the differences in basin geometries and interactions discussed above, edifice-averaged morphometric values
(e.g., Hack's Law exponent, drainage density, mean basin hypsometry, mean basin slopes) are similar to those of
other settings (Hack, 1957; Strahler, 1952; Horton, 1945). This suggests that although radial drainages experience
phenomena that differ from those typically experienced in dendritic settings, drainage development, geometries, and
competition largely follow those of dendritic patterns. As volcanic surfaces are easily datable and their ages can
often vary by orders magnitude on a single edifice, volcanoes thus represent ideal locations for studying terrain
evolution over varying temporal scales within a general framework.

**4.6     Basin morphology capturing volcanic processes**

In this study, we considered edifice morphologies using mean values over the entire edifice. However, our metrics also allow for the comparison of basin morphologies on a single edifice. Variations associated with these metrics would likely relate to spatially-localized attributes of aggradation, degradation, and climate, and would thus provide a quantitative method to disentangle these signals using topography. For example, edifice flanks that have been resurfaced by large volcanic deposits or destroyed by sector collapses should exhibit younger drainage networks according to the metrics explored here, and are expected to differ from other parts of the volcano. Furthermore, alterations to the erosional efficiency of a basin by tephra accumulation or lava flow emplacement should create spatial variability that can be quantified by similar analyses. These concepts should be tested over well-constrained cases and would be beneficial for both preliminary fieldwork and to approximate relative volcanic chronologies remotely. Our model for edifice degradation, radial drainage evolution, and divide stability thus provides a first step to deconvolving the various signals that relate to edifice morphology. This presents new avenues of exploration for the volcanology community to interrogate volcanic histories from topography, and for the geomorphic community to investigate surface evolution on landforms that often fall outside standard tectonic studies.

**5.0  Conclusion**

Volcanic edifices represent a class of primary landforms whose erosion remains relatively unexplored. We analyzed the degradational histories of stratovolcanoes using a set of metrics that have not previously been considered for radial drainage networks. We show that these metrics relate to the overall age of a volcano and propose a new general model for the temporal evolution of edifice drainage morphology. Divide stability analysis underscores the dynamic nature of basin evolution, and suggests that radial drainage networks initiate with nearly-uniform geometries and unstable configurations that evolve towards non-uniform basin geometries and more stable configurations to generate a basin topographic hierarchy on volcanoes. Finally, comparing basin geometries, configurations, and outlet spacing between basins that exist on volcanic edifices to those that exist on linear mountain ranges highlights similarities and differences between radial and dendritic drainage basins.

**6.0  Code availability**

DrainageVolc and MorVolc codes are available at https://github.com/danjohara/Volc_Packages.

**7.0  Data availability**

Collected edifice data is included in the supplement as both an Excel file and shapefile.

**8.0  Author contribution**

All authors provided editorial advice on the manuscript. DO'H wrote the DrainageVolc and updated MorVolc codes, conducted the morphology analyses, and wrote the manuscript. RMJvW assisted in data collection, determined edifice boundaries from topography, and tested DrainageVolc/MorVolc. LG and BC gave advice on drainage basin morphology and evolution, while PG, PL, and GK provided insight on volcanic edifice morphology, evolution, and general volcano ages. MK secured funds and coordinated the project, giving advice on the research direction, analyses, and interpretation.

**9.0  Competing interests**

The authors declare that they have no conflict of interest.

**10.0 Acknowledgement**

This research was funded through the EVoLvE project, Junior FWO project grant G029820N of the Fonds

Wetenschappelijke Onderzoek – Vlaanderen.

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
