# Peer review of "Time-varying drainage basin development and erosion on"

_EGUsphere, 2023_

## Author Comment (AC1)

Dear Susan Conway,

Thank you for consideration of our manuscript for Earth Surface Dynamics. Please find below our responses to reviewer comments (responses are in red italic). We have addressed all suggestions, which has enhanced the scientific quality of the work.

We thank the reviewers for their time, and agree with many of the comments, altering the text appropriately. In particular, we have added a fourth location of volcanoes (Guatemala) to include in our analysis. Incorporating these four volcanoes produce largely similar relationships as before, lending credence to our methodology and interpretation of our results for a generalized model of stratovolcano erosion. The only notable change to our methods was that we now use a drainage area of 1.0 km$^2$ to determine Hack's Law exponent and drainage density. Other modifications to the manuscript include providing further description to our methods and introduction.

Although we do not significantly disagree with reviewer suggestions, we have defended our work over certain points. This includes our use of clustered volcano sets for the edifice morphology analysis, potential bias related to edifice sizes, and the organization of our manuscript.

Despite the small sample size, our choice of analyzing volcano sets has the dominant advantage that they spatially represent varying degrees of edifice degradation. This allows for a simple visual component of studying drainage basin evolution on edifices that would not have been as clear if considering volcanoes that were more spatially scattered. The fact that combining these linear sets together into a single database often generates temporal trends with R$^2$ values > 0.5 suggests to us that our samples are indicative of a general evolution.

In regard to edifice size, we use a consistent method to derive edifice boundaries from topography for all volcanoes that is based on a topographic slope threshold. This methodology is emplaced specifically to remove potential bias in determining the extent of the volcano that is analyzed. Within the volcano sets that were analyzed, some of the volcanoes have smaller areas (e.g., Kaitake has an area of 30 km$^2$, compared to the ~430 km$^2$ of Muria). This was not due to a bias in our boundary designation, but a product of the imposed slope threshold in defining the boundary; and as many of these smaller edifices are often the oldest volcanoes of individual sets, it is reasonable to assume they are representative of later erosional stages and thus highly applicable to our generalized model. Furthermore, the majority of our metrics either implicitly account for edifice size, or are normalized to remove size effects.

Finally, we argue that the organization of the manuscript is appropriate. In particular, the analyses conducted in the discussion are secondary investigations that examine basin evolution and divide migration within the context of our derived model for edifice erosion. They are not the main results of our paper, and thus we suggest they remain within the discussion.

Please let us know if further information is needed, or if there are any questions.

Thank you,

Daniel O'Hara (corresponding author)
Liran Goren, Roos M.J van Wees, Benjamin Campforts, Pablo Grosse, Pierre Lahitte, Gabor Kereszturi, Matthieu Kervyn

**Reviewer #1**

Dear editor and dear authors,

I have read the manuscript entitled "Time-varying drainage basin development and erosion on volcanic edifices" submitted to Earth Surface Dynamics. The manuscript attempted to propose a new conceptual model regarding transition of drainage systems regarding volcanic edifices on the basis of combinations between topographic analysis and ages of volcanic edifices. The topic is well fit for the scope of the journal and the analysis is comprehensive. The idea and analysis are novel, and thus, this study has the potential that improve understanding of landscape evolution in volcanic drainage systems on the basis of geomorphological knowledge accumulated by focusing on non-volcanic fluvial systems. However, to accept wide readers not only geomorphologists but also volcanologists, the authors should address the below my doubts.

1) The authors do not explain reasons why geomorphic and geometrical investigations regarding volcanic edifices have not been carried out well so far. In volcanic edifices, sediment erosion by fluvial processes and sediment supply by eruptions repeatedly occur in various time and spatial scales. Depending on the rate, extent, and pace of these sediment erosion and supply processes, the permeability of volcanic edifices spatiotemporally changes. Moreover, supplied-sediment properties (e.g., grain size and density) have large variations among eruptions even in a single volcano. These mean significant differences in sediment regime of volcanic drainage systems in comparison to general fluvial channels in non-volcanic basins, potentially hampering the direct use of geomorphic and geometric metrics that are developed based on observations in non-volcanic systems. Some metrics used are modified to apply investigations in volcanic edifices, but it seems that the above issues are not completely described in the Introduction section. Additionally, the history of sediment dynamics and eruptions of the respective volcanoes should be well described to support understanding of the current drainage and sediment-transport systems.

*We thank the reviewer for their suggestions. Extra description on the challenges of analyzing volcano basins is now provided in the Introduction (lines 52 – 61). We have stressed specific aspects of drainage patterns as well as a variety of volcano erosional processes in the Introduction; however, as sediment regimes within volcanic fluvial systems are not the focus of this paper, we keep the description general.*

2) A total of 12 volcanoes are investigated but only 9 volcanoes are available to use deriving regressions. Additionally, the authors focused on only closely-spaced sets of volcanic edifices, but its reason is not well described. The presented analysis approach can be applied to other volcanoes. For example, the set of Mt. Sumel, Kawi, and Arujuna which are located in the east of Mt. Merapi that the authors analyzed can be candidates. I struggle to find why the authors did not analyze more volcanoes despite the use of global topographic data. Readers would doubt why the authors excluded the volcanoes in the Cascade Range despite the metric used (eroded volume) was developed based on them in the author's previous study (https://www.frontiersin.org/articles/10.3389/feart.2023.1150760/full). To support the argument

that "a generalized morphologic age of an edifice may be derived that quantifies the erosional state of the landform and relates to the edifice's lithologic age" (L200-202), a wider analysis should be done. Because volcanoes have variations in the rate, extent, and pace of these sediment erosion and supply processes as mentioned above, this lack of enough samples may not be adequate for deriving a new conceptual model.

*Given the reviewer's concerns, we extended our analysis to include four volcanoes from Guatemala with age constraints. This did not significantly alter our results, and relevant sections were updated appropriately.*

*Given the relationships that exist when considering these volcanoes from different arcs, we argue that the trends are representative of stratovolcano erosion. We agree with the reviewer that more data could help refine the model; however, it also risks incorporating volcanoes with specific histories that may obscure the generality presented here, and we are clear in the manuscript that various other processes (e.g., flank collapses, glaciation, and multiple eruption centers) would likely deviate a volcano's morphologic evolution away from our proposed model (lines 296 – 300). The overarching goals of this manuscript were to present the Matlab-based MorVolc and DrainageVolc algorithms, and derive simple morphologic trends that act as a foundation for future studies on edifice erosion. Indeed, a follow-up study is currently underway (led by coauthor R.M.J van Wees) that will test these relationships using an extended database of Indonesian and Japanese stratovolcanoes, as well as the effects of climate and tectonics.*

*The reasoning behind using sets of volcanoes is now made more explicit in the Methods section (lines 79 – 85). One of the overarching reasons to analyze these is that each volcano within a set has had different lifespans of activity over approximately similar climate conditions, giving varying degrees of degradation over a short distance that essentially substitutes space and time, allowing for a visual aid to document changes in morphology. The initial reasoning behind using sets of volcanoes was to analyze the morphologic evolution **within** sets; however, our analysis identified even stronger relations **across** sets and independent of geographic location. Despite this, the manuscript lays the groundwork for future work that can consider regional differences in evolution.*

*Finally, our analysis is focused on the evolution of drainage basins from fluvial erosion, and we thus only consider volcano sets that do not have an extensive glacial history. Although Taranaki has experienced glaciation within the late Holocene (Brook et al., 2011), it is not as extensive as that of Cascades volcanoes (Batchelor et al., 2019). Furthermore, the majority of Cascades stratovolcanoes are isolated and do not experience the same spatial degradational trends observed in these sets. Thus, despite the author's previous work on the Cascades, volcanoes of this arc are excluded from our analysis.*

3) Another concern is bias due to volcano sizes. The size (area) of volcanoes tended to be larger as the value of age of activity (YPB) decreased. Indeed, Kaitake (YPB = 669,627), Pouakai (YPB = 351,994), Ungaran (YPB = 387,298), and Linkruanga (YPB = 160,623) have relatively small areas compared with other volcanoes such as Merapi (YPB = 173). This aspect links with the presented conceptual model in terms of the decreasing elevation of edifices that accompanies a decrease in the corresponding area size. In this context, my doubt is that volcanoes with low

elevations and large areas surrounded by the edifice boundary defined by the authors can exist. If those cannot exist, reliability and robustness of the author's argument and presented model are strengthened.

*We interpret the trend between decreasing volcano area and time to likely relate to the presence and extent of the edifice's sedimentary apron, as opposed to a bias in edifice size. As a volcano erodes, it is expected that material is deposited at the base of the edifice's flank, expanding its areal extent. Given that our edifice boundaries are defined in-part by a 3° slope break between the edifices and surrounding terrain, an inverse relationship between edifice height and area would be expected. Such a relationship would also be expected if one considers a purely diffusive model of edifice erosion. However, we do not observe such a relationship on the 100's of kyr scale that we analyzed. This suggests that sediment is not just depositing at the edifice's base, but is also being evacuated away from the system. Although diffusive processes would eventually spread sediment to slopes < 3°, fluvial transport is more likely the prominent process at the edifices studied here. The expectation from this is that the older edifices (Kaitaki, Pouaki, Ungaran, etc.) may have been as large (in both height and area) as the younger members of their respective set; however, this is of course dictated by the edifice's construction history. We do not dispute that stratovolcanoes could erode to have low relief and large areal extents, and such low aspect values have been cataloged (e.g., Grosse et al., 2014), but further research is needed to determine the climate conditions and constructional histories related to this. This is now added as part of Section 4.1 (lines 290 – 295).*

*Finally, we also note that all other metrics are either size-independent (e.g., Hack's Law exponent, summit basin hypsometry integral, irregularity and ellipticity indices), or are normalized to account for varying sizes (e.g., drainage density, summit basin mean length/width/relief, normalized eroded volume).*

4) In the presented conceptual model (Figure 5), in all ages, volcanoes with a single summit are considered. However, as shown in Figure 9d, in Kaitake (late-stage volcano), the basins drain from the different summits. This may be similar in Pouakai. The fundamental difference regarding the drainage regime is an important aspect, but not considered well. Most volcanic summits would potentially experience collapse in a geological timescale. If so, the related changes in the drainage system can be a general process and also should be touched on in the manuscript. Moreover, the definition of basin outlet is different between early- and late-stage volcanoes. In Merapi (early-stage volcano), because the outlets of summit basins nearly correspond to the edifice boundary, the basins have elongated oval shapes (Figure 9b). In Kaitake (late-stage volcano), the outlets of summit basins are probably located at the outside of the edifice boundary, and consequently, the lower areas of some basins are linearly cut. In another late-stage sample (Likuruanga), the lower ends of basins are constrained by flowing into the sea (Figure 1b). These differences can be responsible for the bias of results in the analysis regarding basin morphology (i.e., Figure 3).

*We showed a single-peaked edifice in Figure 5 for simplicity; however, multiple studies have recognized that older volcanos lose the single peak over time (e.g., Ollier, 1988; Karátson et al., 2012). Although we cannot rule out the possibility of multiple initial summits, we interpret the top of Kaitake to be an erosive feature instead of constructional – as different drainage basins*

*extend upward and start competing around the summit, basin divides become a more complex structure with potentially multiple high points. In this way, we suggest Kaitake is still defined by a single summit, but containing multiple peaks. However, even if Kaitake is interpreted to have multiple summits we would not expect this to significantly change the general model presented here as summits are sufficiently close to each other to still generate a general conical morphology, and Kaitake and Pouakai both fit well within many of the trends from Figures 3 and 4 (e.g., mean summit basin width and length, ellipticity and irregularity indices). We have added a better description of this evolution in Section 4.1 (lines 286 – 290), and have slightly modified Figure 5c to demonstrate a flatter, rougher edifice. This is also mentioned in Section 4.5 (lines 461 – 465).*

*Flank collapses are a different mechanism for edifice erosion, and we discuss their impact throughout the manuscript (lines 312, 428, and 486). Furthermore, we suggest that the deviation of Muria from some of the landform morphology trends in Figure 4 may be explained by its complex history (McBirney et al., 2003), including collapse (line 240).*

*Finally, in reality, all edifice basins extend past our imposed boundary, becoming part of higher-order drainage systems until they eventually reach the ocean. We use the 3° slope threshold to quantify the extent to which we assume the edifice's topography influences basin morphologies; however, other thresholds could have been used. This is consistently applied, and therefore our choice is not likely to introduce bias. It is true that Likuruanga is a special case as its boundary is partially defined by the coast; however, its morphometry relationships do not significantly deviate from the general trends shown in Figure 3 and 4, suggesting this boundary does not produce a bias.*

5) Many contents in subsections 4.2, 4.3, 4.4, and 4.5 should be placed in the methods or results sections. This mixing causes confusion. The manuscript should be reorganized to improve its presentation and coherence.

*After reconsidering the manuscript organization, we prefer to maintain the current structure, as we believe it better reflects the hierarchy and principal importance of our analysis. The main method of our manuscript is the derivation of the DrianageVolc / MorVolc algorithms, and the main results being the temporal trends shown in Figures 3 and 4.*

*Subsections 4.2 – 4.3 (and associated Figures 6 – 8) are secondary, exploratory analyses that build off of the conceptual model presented in Subsection 4.1 and Figure 5, and follow linear narration of the research (i.e., we analyze the morphologies of stratovolcanoes, derive a general model for stratovolcano erosion, then explore how divides migrate and basins evolve within the context of the model). Subsection 4.4 and Figure 9 expands on the main results by exploring what the Hack's Law relationships imply for basin geometries on radial landforms (which has not previously been investigated), as well as where such relationships break down. Subsection 4.5 places our results into the broader context of landscape evolution by comparing edifice basin morphologies with those analyzed in other settings (e.g., linear mountain ranges, fault blocks, passive margins).*

Minor comments

L61 complimenting -> complementing?

*Text has been changed.*

L62 The authors should highlight this study focuses on only stratovolcanoes throughout the manuscript.

*Indication that we are analyzing stratovolcanoes has been added.*

L72 Please explain the reason why only closely-spaced sets were focused.

*Reasoning has now been made explicit (lines 79 – 85).*

L74-75 Please explain the reason for this exclusion.

*Reasoning is now provided (lines 85 – 87)*

L77 Because van Wees et al. (2021) is an abstract rather than a paper, readers cannot follow the method used here.

*Methodology has now been made more explicit (lines 93 – 99).*

L83 More clear and robust explanations are necessary.

*Better description is now provided.*

L86-87 Is this a reason for the volcano selection?

*See reply to Point #2 above.*

L123-125 Please refer to Figure S2 and Table T1 effectively.

*References to figure and table now given.*

L187-188 Eroded volume is not a general metric. Thus, a brief description is required.

*Description of the method is now included (lines 201 – 206).*

**Reviewer #2**

Overall: This is a very interesting paper that creates a solid framework for understanding the interplay between volcano construction and volcano degradation. Surprising results include the co-decrease of height and radius (although height is faster) and the decrease in normalized slope variance with age. The later was surprising because I think of scoria cones (the volcanoes I'm most familiar with) as getting rougher over time (more deep incisions) but understand that the increase in the mean slope is driving the increase in the term.

The most valuable contributions of this work include (a) the port of MorVolc code to Matlab, which is greatly appreciated, and the sharing of that and DrainageVolc through GitHub and (b) the development of a new measurement of slope variance, which seems promising for quantifying the changes in volcanic landforms, large and small. The visual shifts and differences are often apparent in both topography and photos but have not been previously quantified.

My main caveat or disagreement is that I am not sure the mean eruptive age of a volcano is indicative of its landform development state. I would have expected the landform to reflect the last significant eruptive activity.

*We thank the reviewer for their comments. Multiple factors likely influence whether the last significant eruption or mean eruptive age is the most appropriate indication of the landforms state, and this also depends on what metric is being analyzed. For example, edifice height and radius should be based on the total accumulation of material, while many of the drainage metrics should be influenced more by the last major eruption that covered (re-surfaced) the edifice; however, slope is a significant factor in fluvial erosion, so the overall height of the edifice (and thus total volcano output) may also have an impact. Furthermore, 'significant' is an important, but vague, term to quantify – within the context of stratovolcano morphology, is eruption significance based on the amount of erupted material, the percentage of edifice flanks that are permanently covered, a combination of the two, or a different factor altogether?*

*We do not disagree with the reviewer, but suggest this subject is laced with intricacies and needs more discussion within the broader volcanic geomorphology community. Here, we focus on mean eruptive ages as dates of the oldest and youngest known eruptions were reported in the literature for the volcanoes we analyzed, whereas dates of the last major eruption were not. Despite this, the high $R^2$ values of some relationships reported in Figures 3 and 4 (e.g., summit basin hypsometry, summit basin width, edifice height, and main flank ellipticity) suggest these metrics may be used to estimate mean volcano age from morphology.*

*Although not entirely the same as the reviewer's suggestion, we now include a supplement figure (Figure S6) that shows logarithmic regressions between metrics and the age of the last known eruption. Overall regression $R^2$ values are similar to those using the mean activity age, but we note significant decreases in values for mean summit basin length, summit basin width, edifice height, and main flank ellipticity.*

Finally, here are the questions would I like answered in the manuscript: What is the minimum size of volcano considered? Were all volcanoes considered classified (or classifiable) as composite or stratovolcanoes? What are "sinks in the DEM" and why do they need to be filled?

The first two questions concern how applicable these results are to volcanoes in general as opposed to just the larger stratovolcanoes or composite volcanoes that dominate landscapes. The later may be displaying my ignorance of GIS processing (if so, a simple reply rather than manuscript edits would suffice).

*Of the volcanoes we analyzed, edifice sizes (as the planform area of edifice boundaries) ranged from ~30 km$^2$ (Kaitake) to ~433 km$^2$ (Muria); all volcanoes are listed by the Smithsonian Global Volcanism Program (Global Volcanism Program, 2013) as stratovolcanoes. This is now incorporated into the manuscript (lines 87 – 88, 100 – 101). From this, our conclusions should be valid for composite volcanoes with a range of sizes, but radius should be greater than 2-3 km to allow drainages to actually form. Furthermore, we are currently preparing a follow-up manuscript that uses numerical modeling to explore the effects of edifice size on the development of erosion patterns at the scoria cone – stratovolcano transition.*

*A "sink" refers to a DEM pixel that is lower than surrounding cells and thus drives flow paths to it, creating internally drained basins instead of flow to the grid edges. This is different from a morphologic feature of an edifice such as a closed crater (which DrainageVolc is able to adjust for through user parameters), and generally occurs from errors in the DEM. These are therefore often filled through various algorithms (Schwanghart and Scherler, 2014). For the volcanoes we analyzed, there were no indications of closed craters that would have allowed sinks to exist, we thus removed them with a standard fill algorithm.*

References seemed adequate although I'm unfamiliar with the geomorphology literature.

I have no specific line by line comments - the grammar and typography were fine as is and figures great.

**References**
Batchelor, C.L., Margold, M., Krapp, M., Murton, D.K., Dalton, A.S., Gibbard, P.L., Stokes, C.R., Murton, J.B., and Manica, A., 2019, The configuration of Northern Hemisphere ice sheets through the Quaternary: Nature Communications, v. 10, p. 1–10, doi:10.1038/s41467-019-11601-2.

Brook, M.S., Neall, V.E., Stewart, R.B., Dykes, R.C., and Birks, D.L., 2011, Recognition and paleoclimatic implications of late-Holocene glaciation on Mt Taranaki, North Island, New Zealand: Holocene, v. 21, p. 1151–1158, doi:10.1177/0959683611400468.

Global Volcanism Program, 2013, Volcanoes of the World, v. 4.10.5 (27 Jan 2022): Smithsonian Institution,.

Grosse, P., Euillades, P.A., Euillades, L.D., and van Wyk de Vries, B., 2014, A global database of composite volcano morphometry: Bulletin of Volcanology, v. 76, p. 1–16, doi:10.1007/s00445-013-0784-4.

Karátson, D., Telbisz, T., and Wörner, G., 2012, Erosion rates and erosion patterns of Neogene to Quaternary stratovolcanoes in the Western Cordillera of the Central Andes: An SRTM DEM based analysis: Geomorphology, v. 139–140, p. 122–135, doi:10.1016/j.geomorph.2011.10.010.

McBirney, A.R., Serva, L., Guerra, M., and Connor, C.B., 2003, Volcanic and seismic hazards at a proposed nuclear power site in central Java: Journal of Volcanology and Geothermal Research, v. 126, p. 11–30, doi:10.1016/S0377-0273(03)00114-8.

Ollier, C., 1988, Volcanoes (B. Blackwell, Ed.): Oxford:, 288 p.

Schwanghart, W., and Scherler, D., 2014, Short Communication: TopoToolbox 2 - MATLAB-based software for topographic analysis and modeling in Earth surface sciences: Earth Surface Dynamics, v. 2, p. 1–7, doi:10.5194/esurf-2-1-2014.

---

## Editor Decision (ED1)

[revised manuscript text omitted]

---

## Author Response (AR2)

Dear Editors,

Thank you for acceptance of our manuscript to Earth Surface Dynamics. Please find below our responses to your review (as before, responses are in red italic). We have addressed all comments, and agree with the majority of suggestions.

Please let us know if further information is needed, or if there are any questions.

Thank you,

Daniel O'Hara (corresponding author)
Liran Goren, Roos M.J van Wees, Benjamin Campforts, Pablo Grosse, Pierre Lahitte, Gabor Kereszturi, Matthieu Kervyn

Figure 1: Explain hillshaded relief and describe the insets in the caption. Consider using lat/long on the insets (or in caption) so the whitespace can be decreased to make the panels bigger. consider using a single colour-scale for every panel so they can be compared.

*Hillshaded relief, map projection, and inset description has been added to caption. All maps are now given the same color scales. Axes titles have been removed from the inner sections of panels to decrease white space.*

Line 93: Please briefly state what map projection was used and if any distortion associated (if it was an equirectegular then there will be some distortion for NZ)

*Projection description added in both Fig. 1 caption and line 96.*

Line 124: Put citation at end of sentence line 128?

*Citation put at the end of the sentence (now line 130). This was also completed for the Hack's Law equation (line 120).*

Figure 2: Need to explain the projection of the x-y coordinates in metres in the caption (or remove them as the location of the volcano is included on fig 1)

*X-Y coordinates removed from the figure, with a scale bar now placed to give distance.*

Figure 3: Figures 3 and 4, I would like to see some estimates on uncertainty on the vertical points where SD cannot be calculated (to be detailed in the methods)

*After consideration, we disagree with this suggestion to Figs. 3 and 4. The metrics that currently have vertical bars are those that generate a population of values for a single edifice, such that the solid points are the average value and the vertical lines provide the range of values as the standard deviation. This is different from an uncertainty associated with, for example, changing DEM source, altering edifice boundaries, or using a different summit designation. Comparatively, values without vertical bars are singular measurements derived from the edifice's geometry; alterations to their values would thus be associated with the uncertainties previously described. Combining these with the standard deviations presented in Figs. 3 and 4 conflates ideas and would likely lead to confusion.*

*A sensitivity analysis of the uncertainties above on our values is outside the scope of our manuscript, but is the focus of a companion manuscript by co-author R.M.J van Wees* (van Wees et al. in review), *currently in the second stage of reviews in Geomorphology.*

*Although we do not provide the suggested estimations in Figs. 3 – 4, we do now list sources of uncertainty and make this difference clear in the Methods (lines 209 – 215). Furthermore, we have added vertical bars to Irregularities and Ellipticity Indexes in Figs. 4, S5, and S6, which were previously not included but do quantify the standard deviations from an edifice-scale population of samples.*

Axis ticks barely visible and not at all visible on outlined plots

*Axis ticks have been extended to be made more visible.*

The empty circles have dots in them on the plots - is there a significance?

*This was an artifact of Matlab's errobar function and has no significance to the figure. This has been removed here, as well as in Figs. 4, 7, 8, as well as the supplemental figures.*

Figure 4: Axis ticks barely visible and not at all visible on outlined plots

*Axis ticks have been extended to be made more visible.*

Line 264: Evolution to that observed

*Text has been changed (now line 274).*

Figure 5 caption: Analysis of our results?

*Text has been changed.*

Line 290: As the surrounding?

*Text has been changed (now line 301).*

Figure 6: note that background image is hillshade

*Text has been added.*

Suggest making panels bigger and removing whitespace by removing the x,y axes - the location of the volcanoes is given in Fig 1. The xy units are not explained (so if kept the projection system used to generate these coordinates needs to be specified).

*X-Y coordinates removed from the figure, with a scale bar now placed to give distance.*

Figure 7: Delete repeated "log"

*Text has been deleted.*

Figure 9: See earlier comments about the xy coordinates

*X-Y coordinates removed from the figure, with a scale bar now placed to give distance.*

Mention semi-transparent hillshaded relief of SRTM

*Text has been added.*

**Reference**

van Wees, R.M.J., O'Hara, D., Kereszturi, G., Grosse, P., Lahitte, P., Tourniganda, P.-Y., and Kervyn, M. Towards more consistent volcano morphometry datasets: Assessing boundary delineation and DEM impact on geometric and drainage parameters: Geomorphology, in review.